# Melatonin Administration Attenuates High-Fat-Diet-Induced Renal Damage in Wistar Rats

**DOI:** 10.3390/biom16010036

**Published:** 2025-12-25

**Authors:** Olesia Kalmukova, Anastasiia Zavora, Alena Cherezova, Olexiy Savchuk, Mariia Stefanenko, Mykhailo Fedoriuk, Adam C. Jones, Valentyn Nepomnyashchy, Mykola Dzerzhynskyi, Marharyta Semenikhina, Daria V. Ilatovskaya, Oleg Palygin

**Affiliations:** 1Educational and Scientific Center “Institute of Biology and Medicine”, Taras Shevchenko National University of Kyiv, 03022 Kyiv, Ukraine; olesiakalmukova@knu.ua (O.K.); olexiy.savchuk@knu.ua (O.S.); cythistol@knu.ua (M.D.); 2Department of Physiology, Medical College of Georgia, Augusta University, Augusta, GA 30912, USA; anastasiyazavora@gmail.com (A.Z.); acherezova@augusta.edu (A.C.); adajones@augusta.edu (A.C.J.); dilatovskaya@augusta.edu (D.V.I.); 3Academic and Research Medical Institute, Sumy State University, 40000 Sumy, Ukraine; 4Division of Nephrology, Department of Medicine, Medical University of South Carolina, Charleston, SC 29425, USA; stefanen@musc.edu (M.S.); fedoriuk@musc.edu (M.F.); semenikh@musc.edu (M.S.); 5Department of Toxicology, State Institute of Pharmacology and Toxicology, The National Academy of Medical Sciences of Ukraine, 03057 Kyiv, Ukraine; nepoval2@gmail.com

**Keywords:** BMAL1, KIM1, fibrosis, melatonin, circadian disruption

## Abstract

Obesity is a major contributor to kidney injury, in part through circadian rhythms disruption and oxidative stress. Melatonin, a circadian clock regulator, has been proposed as a protective agent against metabolic and renal complications. We investigated the effects of chronic melatonin supplementation on kidney injury and circadian regulation in a rat obesity model. We hypothesized that melatonin administration ameliorates kidney injury induced by a high-calorie diet. Male Wistar rats were fed a normal or hypercaloric diet for six weeks, followed by seven weeks of vehicle or melatonin treatment (30 mg/kg/day in drinking water); biometric parameters and renal injury were assessed. Obese rats exhibited increased visceral adiposity, elevated resistin, renal hypertrophy, fibrosis, tubular degeneration, and glomerular injury, accompanied by higher KIM-1 levels. Melatonin attenuated renal fibrosis, reduced KIM-1, TGFβ, and TNFR1 levels, improved proximal tubule and glomerular damage, and lowered adipose TNF-α levels in the obese groups. In lean controls, melatonin increased nuclear BMAL1 levels, while in obese rats this effect was blunted; of note, BMAL1 accumulated in distal tubular cytoplasm in both melatonin-treated groups. These findings suggest that melatonin mitigates obesity-induced renal pathology through anti-fibrotic inflammation-related mechanisms, while also revealing a novel link between circadian disruption and kidney injury. Our results support melatonin as a therapeutic agent for obesity-related renal disease.

## 1. Introduction

Melatonin and tryptophan metabolism in kidney disease. Melatonin is a circadian-regulating hormone secreted predominantly by the pineal gland during the dark phase of the light cycle, and is recognized for its antioxidant, immunomodulatory, and metabolic regulatory properties [1]. Melatonin is synthesized through the classical indoleamine biosynthesis pathway, which converts tryptophan to serotonin (5-HT) and subsequently to melatonin [2]. It is known that in chronic kidney disease (CKD), tryptophan metabolism is dysregulated, shifting toward the kynurenine degradation pathway, further promoting tissue injury, fibrosis, and inflammation [3,4]. Obesity also shifts tryptophan metabolism towards the kynurenine pathway via activation of indoleamine 2,3-dioxygenase (IDO). On the other hand, blocking the rate-limiting step of tryptophan breakdown by IDO inhibitors to reduce downstream metabolites such as quinolinic acid was proposed as a promising strategy to protect kidney under stress [5]. Of note, melatonin has been shown to attenuate ischemic acute kidney injury (AKI) [6] and diabetic kidney disease [7], and offered nephroprotection in CKD patients [8].

Obesity and circadian rhythm disruption. Obesity is a chronic disease defined by a body mass index (BMI) of 30 or higher, which presents a significant health burden. According to the Centers for Disease Control and Prevention (CDC), the current prevalence of obesity in the US is over 41% [9]. Obesity is a major risk factor for both cardiovascular and renal pathologies, exacerbating hypertension, atherosclerosis, and CKD through complex metabolic, hormonal, and inflammatory pathway dysregulation [10]. The development of many obesity complications is linked to circadian rhythms at both central and peripheral levels, leading to desynchrony in peripheral clock genes, disrupted hormonal secretion patterns, and impaired metabolic homeostasis [11]. These ramifications are primarily attributed to disrupted circadian inputs, including misaligned feeding schedules, such as nocturnal eating in humans, and disturbances in the light–dark cycle due to reduced daytime physical activity and increased exposure to artificial light at night [12].

Melatonin is a circadian regulator of metabolism. Melatonin plays a key role in regulating circadian rhythms [13], and its action in the peripheral tissues is increasingly recognized [14,15,16]. Melatonin acts through its receptors, MT1 and MT2, in the central nervous system (CNS) to synchronize peripheral clocks with the central rhythm [17]. In addition to this canonical pathway, melatonin has been shown to “reset” disrupted circadian rhythms, restoring healthy temporal organization [18]. Emerging evidence also suggests the CNS-independent chronobiological role of melatonin [19]. Notably, melatonin receptors are widely expressed in different tissues (Figure 1). In humans, the kidney is among the tissues exhibiting particularly high expression of the MT1 receptor (Figure 1A), suggesting an important role for melatonin signaling in renal circadian regulation and pathophysiology. Consistent with this observation, rodent kidney expresses both *Mtnr1a* and *Mtnr1b*, encoding MT1 and MT2, respectively. Data from the Susztaklab Kidney Biobank [20] indicate robust *Mtnr1a* expression across multiple renal cell types in rats, while *Mtnr1b* is detectable at lower levels, predominantly in tubular epithelial cells. Similarly to humans, in the rat kidney, MT1 abundance is significantly higher than MT2. In addition, the Staruschenko Lab Hypertension Atlas [21], which reports RNA-seq data from the rat kidney, heart, and liver, detected expression of both melatonin receptors exclusively in the kidney, but not in the liver or heart, which underscores a higher prevalence of melatonin receptor expression in renal tissue compared to other visceral organs, highlighting the kidney as a key peripheral target of melatonin-dependent circadian regulation.

Melatonin is a key regulator of circadian biology through its interaction with core clock genes, including *Bmal1* (Basic helix–loop–helix ARNT-like protein 1) [22]. *Bmal1* is a critical transcription factor in circadian machinery, influencing metabolic pathways such as glucose homeostasis, lipid metabolism, and inflammation [23]. Knockout of *Bmal1* and dysregulation of its expression have been linked to obesity and its associated complications, including insulin resistance and renal dysfunction [24,25,26].

Melatonin receptors play a role in the normal circadian dipping of blood pressure. In patients with essential hypertension, repeated nighttime melatonin intake significantly reduced nocturnal blood pressure and enhanced circadian rhythm amplitude, supporting its role in restoring circadian cardiovascular regulation [27]. Low melatonin levels have been linked to the “non-dipping” phenotype in hypertensive patients, who often experience abnormally low nocturnal melatonin secretion [28]. Moreover, prolonged administration of melatonin in women improved the blood pressure dipping [29]. Finally, studies utilizing an MT1-MT2 dual antagonist, agomelatine, which is a potent agonist of melatonin receptors and a relatively weak serotonin receptor antagonist, showed that melatonin receptor activation in the kidney reduces oxidative stress [30]. The beneficial effects of melatonin receptor activation in various renal pathologies indicate that melatonin signaling plays a key role in regulating renal function. These findings highlight the need for further investigation into renal-specific mechanisms regulated by melatonin.

Mechanisms of melatonin action in renal tissue. In patients with CKD, melatonin secretion is decreased [31], highlighting a potential role for melatonin in maintaining renal homeostasis. Experimental studies demonstrate that melatonin supplementation ameliorates renal dysregulation of the renin–angiotensin–aldosterone system (RAAS) [32,33] and has direct renoprotective effects, including suppression of the cGAS–STING inflammatory signaling pathway in mouse models of acute kidney injury (AKI) [34]. Cherngwelling et al. showed that in a model of obesity-mediated renal injury, agomelatine treatment inhibited endoplasmic reticulum stress and apoptosis [35]. Treatment with another melatonin receptor agonist, tazimelteon, exhibited renoprotection in acute kidney injury [36]. Melatonin receptor agonist ramelteon, which is currently FDA-approved for insomnia, was shown to attenuate renal ischemia and reperfusion injury through reducing mitochondrial fission and fusion [37]. Beyond these signaling pathways, melatonin upregulates BMAL1 expression and activity, thereby generally improving the maintenance of metabolic homeostasis [38]. Exogenous melatonin restores BMAL1 rhythmicity, reduces oxidative stress, and improves mitochondrial function, ultimately attenuating obesity-induced tissue injury [39]. These findings suggest that melatonin’s regulatory effect on BMAL1 may be a key mechanism in its protective actions against obesity-related renal injury and systemic metabolic dysfunction. In this study, we tested the hypothesis that melatonin administration improves obesity-disrupted renal circadian rhythms, ameliorating kidney injury induced by a high-calorie diet.

**Figure 1 biomolecules-16-00036-f001:**
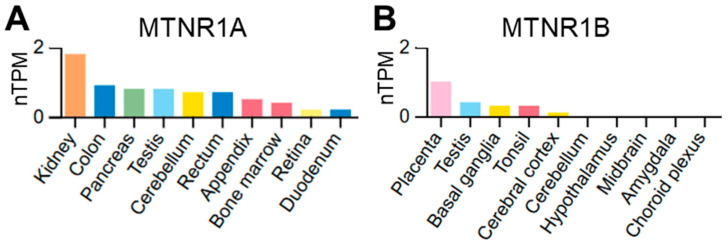
Melatonin receptors expression in human tissue. Bulk tissue expression data for MT1 (MTNR1A, (**A**)) and MT2 (MTNR1B, (**B**)) obtained from the Human Protein Atlas (proteinatlas.org [40]) Consensus dataset. Normalized expression (nTPM) levels were calculated for 55 tissue types (top 10 expression level tissues are shown) by combining the HPA and GTEx transcriptomics datasets. MTNR1A: https://www.proteinatlas.org/ENSG00000168412-MTNR1A/tissue (accessed on 21 December 2025); MTNR1B: https://www.proteinatlas.org/ENSG00000134640-MTNR1B/tissue (accessed on 21 December 2025).

## 2. Materials and Methods

All procedures involving animals were approved by the Ethical Committee of the Educational and Scientific Centre “Institute of Biology and Medicine,” Taras Shevchenko National University of Kyiv, Ukraine. The experimental protocol complied with the Directive 2010/63/EU of the European Parliament and Article 26 of the Law of Ukraine “On the Protection of Animals from Cruel Treatment” and ARRIVE guidelines [41].

High-Fat Diet-Induced Obesity Model and Study Design. Wistar male rats (10 weeks old, initial body weight: 110 ± 10 g) were used for this study. Following a 7-day acclimatization period under standard laboratory conditions (temperature: 22 ± 3 °C; relative humidity: 60 ± 5%; 12:12 h light–dark cycle), the animals were randomly assigned to several groups. The control group (CTRL) received standard rodent chow (Purina, VITA, Obukhiv, Ukraine; energy content: 3.81 kcal/g, 6.7% fat, 21% protein, 55.1% carb) and water ad libitum. The high-fat diet (obesity, OB) group received a hypercaloric (5.35 kcal/g, 38.8% fat, 15.5% protein, 45.7% carb) modified Purina diet, composed of 60% standard rodent chow, 10% pork visceral fat, 10% chicken eggs, 9% beet sugar, 5% raw peanuts, 5% powdered whole milk (26% fat), 1% sunflower oil, and water. Following 6 weeks of dietary protocol, the average weight gain in the OB group exceeded that of the CTRL by ≥30%. Next, animals were further randomized to receive vehicle or melatonin (+ML) in drinking water for the next 7 weeks. Drinking water was provided ad libitum (water consumption among groups was 38 vs. 31 mL/day for lean vs. obese, respectively), yielding a 30 mg/kg/day dose of melatonin (Figure 2).

Food and water consumption were recorded daily between 9:00 and 10:00 a.m. Relative daily food intake (kcal/day/g body weight) and water intake (mL/day/rat) were noted. Rats were weighed weekly to determine body weight gain (%). All animals were euthanized by decapitation at 10:00 a.m. during the light phase of a 12:12 h light–dark cycle. Each day, one group of five animals was sacrificed. Visceral adipose tissues (epididymal, retroperitoneal, and perirenal) and kidneys were carefully dissected, weighed, and divided into portions designated for biochemical analyses (including assessment of resistin levels) and histological evaluation. Relative mass of adipose tissue was calculated as Adipose Tissue mass/Rat mass × 100%. Renal BMAL1 protein level was examined at a fixed time point within the light phase, corresponding to the peak of BMAL1 mRNA expression [42]. This experimental design ensured temporal consistency across groups and minimized potential confounding by circadian fluctuations.

ELISA for resistin and TNFα quantification in adipose tissue. The levels of resistin (adipokine produced by fat tissue and immune cells) were quantified in adipose tissue homogenates using an indirect ELISA performed in 96-well high-binding microplates designed for the adsorption of soluble proteins (M4686, Greiner Bio-One, GmbH, Frickenhausen, Germany). Prior to analysis, serum samples were diluted 1:10 with 50 mM Tris-HCl buffer (pH 7.4) containing 150 mM NaCl. Aliquots of 100 μL of the diluted serum were added to each well and incubated overnight at 4 °C to allow antigen adsorption. Following incubation, wells were washed with 50 mM Tris-HCl (pH 7.4) containing 150 mM NaCl and 0.05% Tween-20 to remove unbound proteins. Nonspecific binding was blocked by incubating wells with 5% skim milk solution at 37 °C for 1 h. After blocking, wells were washed, and primary goat polyclonal anti-resistin antibodies (sc-16117, 1:1000 dilution, Santa Cruz Biotechnology, Dallas, TX, USA) were added and incubated at 37 °C for 1 h. Subsequently, wells were washed again, and secondary horseradish peroxidase (HRP)-conjugated bovine anti-goat antibodies (sc-2350, 1:5000 dilution, Santa Cruz Biotechnology, Dallas, TX, USA) were applied and incubated for an additional hour at 37 °C. Detection was performed using 100 μL of 0.4 mg/mL diaminobenzidine (DAB) prepared in a citrate buffer (pH 5.0) with 0.013% hydrogen peroxide (H_2_O_2_). The enzymatic reaction was terminated after 10 min by adding 100 μL of 1 M sulfuric acid (H_2_SO_4_) to each well. Absorbance was measured at 492 nm using a microplate reader (BioTek Instruments, Winooski, VT, USA). Adipokine concentrations were normalized to total serum protein content, which was determined using the Bradford assay [43].

Prior to TNFα determination, aliquots of adipose tissue homogenate were diluted with 0.05 M Tris-HCl buffer (pH 7.4) to a final protein concentration of 20 μg/mL. TNFα concentration was measured using an enzyme-linked immunosorbent assay (ELISA) following the manufacturer’s recommendations with standard optimization steps. High-binding 96-well plates were coated with antigen (10 μg/mL in 0.1 M NaHCO_3_, pH 9.6) and incubated overnight at 4 °C. The next day, plates were washed three times with Tris-buffered saline (TBS) containing 0.05% Tween-20 and then with TBS without detergent, followed by blocking for 1 h at 37 °C with 1% bovine serum albumin (BSA) in TBS to prevent non-specific binding. After washing, primary antibodies (TNFα, sc-374433, Santa Cruz Biotechnology, Dallas, TX, USA) diluted in TBS according to the manufacturer’s instructions were added and incubated for 1 h at 37 °C. After an additional wash step, HRP-conjugated goat anti-mouse IgG secondary antibodies (sc-2005, Santa Cruz Biotechnology, Dallas, TX, USA) were added and incubated under the same conditions. Detection was performed using o-phenylenediamine dihydrochloride (OPD) substrate in 0.05 M phosphate-citrate buffer containing 0.3% H_2_O_2_. The reaction was allowed to proceed for 10 min and then stopped by adding 2.5 N H_2_SO_4_. Absorbance was measured at 492 nm using a μQuant spectrophotometer (BioTek Instruments, Winooski, VT, USA).

Histological Analysis. For histological evaluation, kidneys were fixed in 4% neutral buffered paraformaldehyde (Sigma-Aldrich, Merck, Darmstadt, Germany) for 72 h. Visceral adipose tissue sections (5 μm) were prepared from paraffin-embedded blocks and stained with hematoxylin (Merck, Germany) and eosin (HLR, Kyiv, Ukraine). All slides were observed by a Leica Aperio Versa 200 digital pathology scanner with an autoloader slide and Leica DM6 B fully automated upright microscope system, producing digital micrographs using a Leica DFC7000 T camera and the Aperio ImageScope 12.3 software (Leica Biosystems Imaging, GmbH, Nussloch, Germany). Fibrosis level in kidney tissue was evaluated using Picrosirius Red staining. The area occupied by collagen fibers (red-stained regions) was quantified as a percentage of the total tissue area using image analysis software FIJI (ImageJ 1.54p, ImageJ, NIH, Bethesda, MD, USA). Proximal tubule (PT) injury was analyzed using periodic acid–Schiff (PAS)-stained slides. Proximal tubules were blindly scored on a 0–4 scale [44]. A score of 0 represented a healthy tubule with intact brush border, no epithelial cell flattening or tubular dilation, regular columnar epithelial structure, and no paracellular gaps; mild tubular degeneration (score 1–2) manifested by epithelial cell atrophy, loss of brush borders, and cell polarity; score of 4 represented 76–100% of tubular necrosis, loss of brush border, protein cast formation, tubular dilation, tubular atrophy, and thickening of the basement membrane (0, 1 score—0–25%; 2 score—26–50%; 3 score—51–75%; 4 score—76–100%). Overall, scores of 0–1 are considered within the range of normal physiology, whereas scores 3–4 indicate the presence of meaningful tubular injury (Appendix A). Glomerular damage was detected on Masson’s trichrome-stained slides. A glomerular injury cumulative score was assessed using morphometric analysis on a 0–4 scale based on the signs of early focal segmental glomerulosclerosis (FSGS), capsule hypertrophy, segmental mesangial, and endocapillary hypercellularity, as described previously [45] (Appendix A).

Immunohistochemistry for Kim1, TGF-β, and Bmal1 detection. BMAL1 primary antibody was incubated with the slide overnight at 4 °C (rabbit monoclonal antibody at a dilution of 1:200; Cell Signaling Technology Danvers, MA, USA, catalog # 14020), and then visualized with horseradish peroxidase coupled secondary antibodies (Rabbit HRP-polymer, Cat # RHRP520, BioCare Medical, Pacheco, CA, USA) and 3,3-diaminobenzidine (Vector Labs, Newark, CA, USA), as was previously reported [46]. KIM1 primary antibody was incubated with the slide overnight at 4 °C (goat polyclonal antibody at a dilution of 1:100; R&D Systems, Minneapolis, MN, catalog #AF3689), and then visualized with horseradish peroxidase-coupled secondary antibodies (ImmPress HRP Anti-Goat IgG polymer, Cat # MP-7405, Vector Labs, Newark, CA, USA) and 3,3-diaminobenzidine (Vector Labs, Newark, CA, USA) [47]. TNFR1 polyclonal antibody (Thermofisher, Waltham, MA, USA, PA5-95585; host: rabbit) was incubated with the slide overnight at 4 °C at a dilution of 1:100, and then visualized with goat anti-rabbit secondary IgG (goat anti-rabbit ImmPRESS HRP kit, Vector Laboratories, Newark, CA, USA, MP-7451) and ImmPACT DAB (Vector Laboratories, Newark, CA, USA, SK-4105). Slides were counterstained for 1 min with hematoxylin. Images from BMAL1 and KIM1 stained slides were taken at 20× using a Nikon Ti-2 microscope equipped with NIS-Elements AR software (NIS-Elements AR version 5.20.02, Nikon, Tokyo, Japan). In total, 17 to 25 images were taken per slide. Then, images were loaded to FIJI (ImageJ 1.54p, ImageJ, NIH, USA) and batch-processed for color deconvolution using Macro plugin and H-DAB staining for intensity (mean gray value, MGV) [48]. For further analysis of BMAL1 levels in distal tubules (DTs), regions of interest (ROIs) were delineated separately for the cytoplasm and nucleus, followed by calculation of the nuclear-to-cytoplasmic ratio (NCR) to assess BMAL1 subcellular distribution and translocation [49]. Formalin-fixed, paraffin-embedded rat kidney sections were deparaffinized, rehydrated, and subjected to heat-induced antigen retrieval, followed by overnight incubation at 4 °C with rabbit monoclonal anti-TGF-β1 antibody (EPR21143, Abcam, Cambridge, UK, ab215715; 1:100). Sections were incubated with goat anti-rabbit IgG (H + L) Alexa Fluor 647 secondary antibody (Invitrogen, Carlsbad, CA, A-21245; 1:1000). Images were acquired using an Evident FV4000 confocal microscope (Evident Scientific, Tokyo, Japan) with acquisition parameters held constant for all samples.

Statistical analysis. OriginPro 2023b and IBM SPSS Statistics v31 software were used for data analyses. Results are expressed as means ± SE (box plot), and whiskers show SD. A minimum of five animals per group were included in all datasets. Two-way ANOVA with Holm–Sidak post hoc multiple comparison tests served for the assessment of significance of the observed changes. A *p* value < 0.05 was considered statistically significant. For the ordinal data of renal glomerular and tubular analysis (cumulative probability damage score), due to the discrete/categorical and non-continuous nature of the data, group differences were assessed using a non-parametric Kruskal–Wallis test with Dunn’s procedure and complementary ordinal association quantified by Cramer’s V. A strength of difference in the range 0.3–0.5 was considered moderate, and >0.50 was strong.

## 3. Results

Biometric parameters. Melatonin administration in obese rats resulted in a significant reduction in body weight gain beginning two weeks after treatment initiation, despite continued consumption of a high-fat diet (Figure 3A). Weekly averaged caloric (Figure 3B) and water intake (Figure 3C), monitored daily throughout the study, confirmed the expected higher calorie consumption in obese animals compared with lean controls. Importantly, melatonin treatment did not alter caloric or water intake within either lean or obese groups, indicating that the reduction in body weight gain in obese rats was not attributable to changes in dietary intake. Histological evaluation of visceral adipose tissue (Figure 3D) revealed abundant crown-like clusters of macrophages surrounding necrotic adipocytes in the obese group, whereas melatonin-treated obese rats displayed fewer inflammatory lesions. Beige adipocytes, indicated by arrows, were readily observed in melatonin-treated lean animals; in the melatonin-treated obese group, they appeared only occasionally and formed smaller clusters, while they were absent in untreated obese rats. Consistent with this, circulating levels of the pro-inflammatory adipokine resistin decreased to values comparable to controls following melatonin treatment (Figure 3E). Melatonin treatment was further associated with a reduction in relative visceral adipose tissue mass in obese animals (Figure 3F), supporting an overall attenuation of adipose accumulation.

Renal damage and inflammatory parameters. Obesity-associated renal hypertrophy (Figure 4A) was reduced in the OB + ML group, with the kidney-to-body weight ratio no longer differing significantly from the control group. Similarly, melatonin treatment reduced overall kidney weight, bringing values closer to controls (1950 ± 86, 2136 ± 162, 2710 ± 114, and 2307 ± 107 mg for CTRL, CTRL + ML, OB, and OB + ML, respectively). Obesity, characterized by low-grade chronic inflammation and tissue hypoxia due to renal hypertrophy, is commonly associated with increased fibrotic remodeling. To assess inflammatory signaling, we performed TNFR1 immunohistochemistry (Figure 4B), with representative images shown for all experimental groups. Quantification demonstrated the highest TNFR1 abundance in obese rats, with a significant decline following melatonin treatment in the OB + ML group (Figure 4C). TNF-α levels in adipose tissue mirrored these findings, exhibiting the same group-dependent pattern and supporting a systemic inflammatory signature that extends to the kidney (Figure 4D). TNFR1, the primary receptor for TNF-α, mediates pro-inflammatory, apoptotic, and necroptotic pathways and is strongly upregulated in states of metabolic stress, including obesity. Consistent with this inflammatory activation, representative Picro-Sirius Red-stained kidney sections illustrate collagen deposition indicative of fibrosis (Figure 4E). Statistical analysis revealed a significant increase in fibrosis in the obese group, which was absent in the OB + ML group, further supporting melatonin’s protective effect on inflammation-driven renal remodeling (Figure 4F).

Proximal tubular (PT) pathology. Representative PAS-stained kidney sections (Figure 5A) from rats reveal marked degenerative changes in obese animals, including intracytoplasmic hyaline droplets (arrows), hyaline casts within tubular lumens, and focal tubular epithelial cell death. Quantitative assessment of proximal tubule (PT) damage was performed using a cumulative scoring system (scale 0–4) across experimental groups (of tubular necrosis, loss of brush border, protein cast formation, tubular dilation, and tubular atrophy). Proximal tubular injury in obese rats was characterized by degenerative changes, accompanied by vacuolization and desquamation of tubular epithelial cells.

The cumulative distribution of tubular damage scores (the percentage of a specific score for each score group indicated on the graph) revealed marked pathology in the OB group, with values shifted toward higher scores in most tubules (Figure 5B). Melatonin treatment (OB + ML group) shows a non-significant trend in tubular damage score reduction and absence of severely injured tubules, although pathology remained greater compared to controls. Kidney injury molecule-1 (KIM-1) immunostaining further confirmed the presence of this injury-associated molecule in PT of obese rats (Figure 5C). Quantitative analysis demonstrated a significant upregulation of KIM-1 abundance in the OB group, consistent with tubular injury. Melatonin treatment (OB + ML group) markedly reduced KIM-1 levels; however, levels remained higher than those observed in controls (Figure 5D). Consistent with the observed PT injury, immunofluorescence analysis revealed a marked increase in TGFβ abundance in the renal cortex of obese rats (Figure 5E), a key profibrotic cytokine implicated in tubular epithelial injury and the progression of tubulointerstitial fibrosis. Quantitative analysis confirmed that obesity significantly increased TGFβ levels (Figure 5F), supporting the activation of fibrotic signaling pathways in parallel with PT damage. Melatonin treatment attenuated this obesity-induced increase in TGFβ abundance, indicating partial suppression of profibrotic signaling, although levels did not fully normalize to control values. Together, these findings link obesity-driven upregulation of TGFβ with proximal tubular injury and suggest that melatonin-mediated renoprotection involves, at least in part, the modulation of TGFβ-associated fibrotic responses. Overall, these findings support a partial renoprotective effect of melatonin in mitigating obesity-induced tubular injury.

Glomerular damage. To further assess glomerular injury, we evaluated Masson’s Trichrome-stained kidney sections (Figure 6A). The scoring criteria included early focal segmental glomerulosclerosis (FSGS), segmental mesangial hypercellularity, capsular hypertrophy, mesangial matrix expansion, and periglomerular fibrosis. Obese rats displayed a significantly higher cumulative glomerular damage score, with 38% of glomeruli classified as severely affected (Figure 6B). Melatonin treatment (OB + ML group) reduced glomerular injury, with only 9% of glomeruli exhibiting high damage scores. Notably, statistical analysis indicated that the strength of group differences for glomerular injury was moderate compared to the more pronounced tubular alterations. Whole-kidney hypertrophy in the OB group (Figure 4A) was also reflected in a significant increase in glomerular cross-sectional area, which was attenuated by melatonin administration (Figure 6C).

Renal BMAL1 level changes. BMAL1, a core circadian clock transcription factor, was examined in kidney tissues to assess circadian regulation at the molecular level. BMAL1-stained kidney sections and deconvoluted images were analyzed for staining intensity using the mean gray value (MGV) parameter (Figure 7A). As previously reported [46], we observed prominent nuclear BMAL1 abundance in renal tubules, which was further increased in the CTRL + ML group following melatonin treatment. Interestingly, melatonin administration also induced cytoplasmic localization of BMAL1 in distal tubule cells across both control and obese groups (Figure 7B). Quantitative analysis revealed a robust increase in nuclear BMAL1 levels in response to melatonin in controls, whereas this effect was absent in obese rats (Figure 7C). In contrast, distal tubular abundance showed a similar trend of elevation in both melatonin-treated groups, reflecting the enhanced cytosolic localization of BMAL1 (Figure 7D). Overall, these findings suggest that while obesity alone did not significantly alter BMAL1 levels compared to controls, melatonin modulated BMAL1 localization in a context-dependent manner. In healthy kidneys, melatonin enhanced nuclear BMAL1 accumulation and increased its cytosolic presence in distal tubules. In contrast, under obese conditions, melatonin shifted BMAL1 distribution primarily toward distal tubules, accompanied by a marked reduction in nuclear staining. This highlights the possibility that melatonin exerts tissue- and compartment-specific effects on renal circadian regulation.

## 4. Discussion

This study demonstrates that chronic melatonin administration attenuates high-fat-diet-induced renal injury in rats, with measurable improvements in tubulointerstitial damage, glomerular pathology, fibrotic remodeling, and inflammatory signaling. By integrating histopathologic, molecular, and metabolic readouts, our findings provide a data-driven framework linking obesity-associated kidney injury with melatonin-sensitive pathways, while also delineating the boundaries of what can be concluded regarding inflammation and circadian regulation.

A central finding of this study is that obesity induces prominent proximal tubular injury, characterized by epithelial degeneration, tubular dilation, cast formation, and increased abundance of KIM-1, a well-established marker of tubular damage. This was accompanied by increased renal TGFβ abundance, supporting the activation of profibrotic signaling pathways that are known to couple tubular epithelial injury with progressive tubulointerstitial fibrosis [50,51]. Melatonin treatment partially mitigated these effects, reducing KIM-1 levels and attenuating TGFβ accumulation, although neither parameter fully normalized to lean control levels (Figure 5). These findings indicate that melatonin exerts partial but biologically meaningful renoprotection, consistent with a disease-modifying rather than fully restorative effect in established obesity-induced kidney injury.

Beyond tubular pathology, obesity also promoted renal hypertrophy, increased collagen deposition, and glomerular injury, including mesangial expansion and early focal segmental glomerulosclerosis-like features. Melatonin significantly reduced renal fibrosis and glomerular enlargement, supporting a coordinated protective effect across multiple renal compartments. Importantly, the magnitude of melatonin’s effect was more pronounced in tubular and interstitial indices than in glomerular injury scores, suggesting that tubulointerstitial compartments may represent a primary target of melatonin-mediated protection in obesity.

To address inflammatory mechanisms, we expanded the analysis to include TNFR1 abundance in kidney tissue and TNFα levels in adipose tissue, allowing a more direct assessment of inflammatory signaling than resistin alone. Obese rats exhibited increased TNFR1 abundance in the kidney, consistent with the activation of TNFα-TNFR1 signaling pathways that promote tubular injury, apoptosis, and fibrotic remodeling [52,53]. Melatonin treatment reduced renal TNFR1 and adipose TNFα levels, supporting the attenuation of obesity-related inflammatory signaling. However, in the absence of a broader renal inflammatory cytokine panel or immune cell profiling, these findings should be interpreted as evidence of involvement of selected inflammatory pathways, rather than a generalized anti-inflammatory effect. Accordingly, our data support the conclusion that melatonin dampens TNFα/TNFR1-associated inflammatory stress, which likely contributes to its renoprotective actions.

Circadian disruptions have been implicated in the pathogenesis of renal diseases and are increasingly recognized as therapeutic targets [54]. This study also explored the relationship between melatonin treatment, obesity-induced kidney injury, and circadian clock-related mechanisms, focusing on BMAL1 as a core molecular clock component. We observed that melatonin increased nuclear BMAL1 abundance in lean control kidneys, consistent with prior reports demonstrating melatonin-mediated enhancement of circadian clock function [22,55]. In contrast, this nuclear response was blunted in obese animals, suggesting impaired circadian responsiveness in the obese kidney. Notably, melatonin induced a redistribution of BMAL1 toward the cytoplasm of distal tubular cells in both lean and obese groups, indicating segment-specific and context-dependent effects of melatonin on BMAL1 localization. These findings do not establish the restoration of circadian rhythmicity per se, as BMAL1 was assessed at a single time point and rhythmic oscillations were not measured. Instead, the data suggest that obesity alters the renal cellular response to melatonin at the level of BMAL1 regulation, potentially uncoupling melatonin signaling from canonical nuclear clock function while preserving non-canonical or compartment-specific actions in distal tubules. Given emerging evidence that circadian disruption exacerbates renal injury and fibrosis, altered BMAL1 localization may represent a mechanistic link between metabolic stress and renal vulnerability. Nevertheless, future studies incorporating time-series analyses and functional circadian outputs will be required to define the precise role of circadian dysregulation.

Taken together, our results support a model in which high-fat-diet-induced obesity promotes renal inflammation, profibrotic signaling, and tubular injury, while melatonin attenuates these processes through combined metabolic, anti-fibrotic, and inflammation-related effects. The kidneys are involved in melatonin catabolism and are among the tissues with the highest melatonin accumulation [48]; thus, renal cells are highly exposed to melatonin and its metabolites. The renoprotective actions of melatonin appear to involve the suppression of TNFR1-associated stress and partial inhibition of TGFβ-driven fibrotic pathways, alongside context-dependent modulation of BMAL1 distribution within renal tubules.

## 5. Conclusions

In conclusion, this study demonstrates that melatonin administration mitigates obesity-induced renal injury, particularly within the tubulointerstitial compartment, by reducing tubular damage, fibrotic remodeling, and TNFα/TNFR1-associated inflammatory signaling. While melatonin modulates BMAL1 abundance and localization in renal tissue, our findings indicate altered circadian responsiveness in obesity rather than full restoration of circadian rhythmicity. Together, these data position melatonin as a disease-modifying agent in obesity-related kidney injury, acting through anti-fibrotic and inflammatory pathway modulation, with potential contributions from circadian clock-related mechanisms. Further studies incorporating temporal analyses and expanded inflammatory profiling will be essential to define the full scope of melatonin’s renoprotective actions.

## Figures and Tables

**Figure 2 biomolecules-16-00036-f002:**
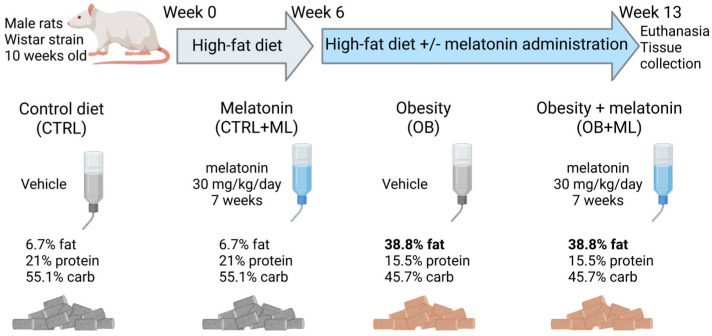
Experimental design. Male Wistar rats at 10 weeks of age were randomly assigned to (1) control group (CTRL), which received standard rodent Purina chow and water ad libitum), or (2) high-fat diet (obesity, OB) group, which received a hypercaloric Purina chow for 6 weeks. Next, animals were further randomized to receive vehicle or melatonin (ML) at 30 mg/kg/day in drinking water for 7 additional weeks, resulting in the following experimental groups: CTRL, OB, CTRL + ML, and OB + ML (n ≥ 7 animals per group).

**Figure 3 biomolecules-16-00036-f003:**
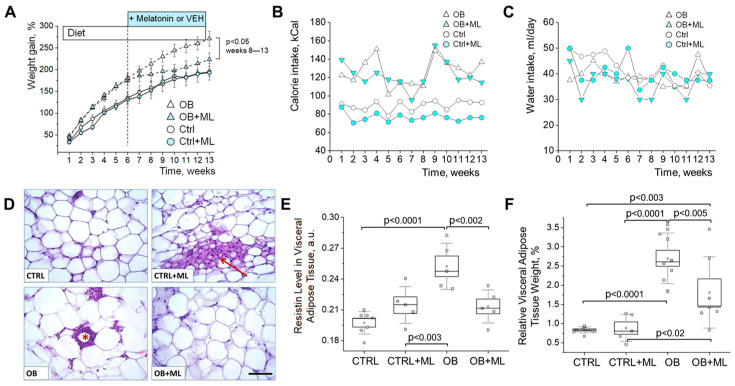
Melatonin treatment attenuates obesity development and adipose tissue accumulation. (**A**) Dynamics of weight gain change during the experiment timeline (*p* < 0.05, OB vs. OB + ML; *t*-test). (**B**) Average daily caloric intake for the week (N ≥ 5 rats per group) (**C**) Average daily water intake for the week (N ≥ 5 rats per group). (**D**) Representative H&E-stained visceral adipose tissue sections (scale bar: 100 μm) showing a “crown”-like structure marked with an asterisk representing dead adipocytes surrounded by macrophages; beige adipocytes are marked with an arrow. (**E**) Quantification of resistin level in visceral adipose tissue (ELISA, N ≥ 5 rats per group, two-way ANOVA with Holm–Sidak; weight (factor1): *p* < 0.001; treatment (factor2): ns). (**F**) Quantification of relative adipose tissue weight (N ≥ 5 rats per group, two-way ANOVA with Holm–Sidak; weight (factor1): *p* < 0.001; treatment (factor2): ns). CTRL—control group; CTRL + ML—lean rats with melatonin administration; OB—obese group; OB + ML—obese rats with melatonin administration; ns indicates no statistically significant difference.

**Figure 4 biomolecules-16-00036-f004:**
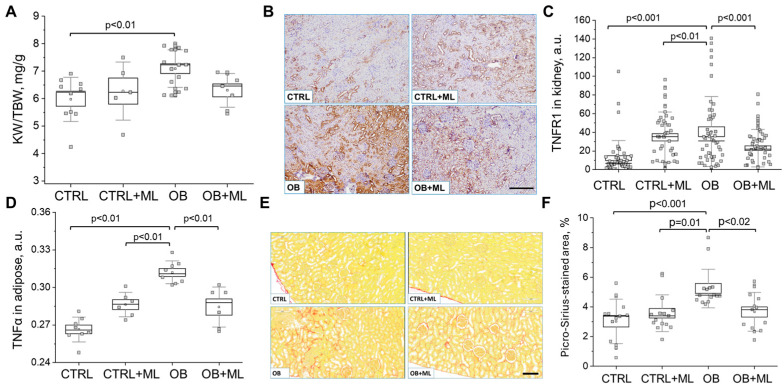
Melatonin treatment ameliorates renal fibrosis. (**A**) Kidney weight to body weight (N ≥ 5 rats per group, two-way ANOVA with Holm–Sidak; weight (factor1): *p* < 0.02; treatment (factor2): ns). (**B**) Representative TNFR1 (Tumor Necrosis Factor Receptor 1)-stained kidney sections from rats (scale bar: 300 μm). (**C**) Quantification of TNFR1 abundance using image color deconvolution (n ≥ 50 slides/group (5 rat/group), two-way ANOVA with Holm–Sidak; weight (factor1): *p* < 0.03; treatment (factor2): ns). (**D**) Quantification of TNFα (Tumor necrosis factor-alpha) level in visceral adipose tissue (ELISA, N ≥ 5 rats per group, two-way ANOVA with Holm–Sidak; weight (factor1): *p* < 0.001; treatment (factor2): ns). (**E**) Representative Picro-Sirius-stained kidney sections (scale bar is 200 μm). (**F**) Quantification of fibrosis level as relative area (n = 15 slides/group (3 slides per rat), two-way ANOVA with Holm–Sidak; weight (factor1): *p* < 0.02; treatment (factor2): ns). CTRL—control group; CTRL + ML—lean rats with melatonin administration; OB—obese group; OB + ML—obese rats with melatonin administration; ns indicates no statistically significant difference.

**Figure 5 biomolecules-16-00036-f005:**
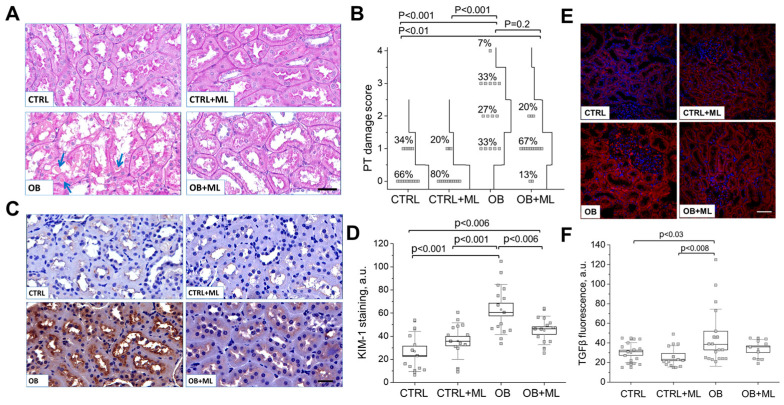
Melatonin treatment reduces tubular damage. (**A**) Assessment of proximal tubular (PT) damage. Representative PAS-stained kidney sections from rats (scale bar: 50 μm). Obese rats exhibit prominent degenerative changes, including intracytoplasmic hyaline droplets (arrow), hyaline casts within tubular lumens, and focal tubular epithelial cell death. (**B**) Quantification of PT damage scores (scale 0–4) across control and experimental groups. A score of 0 denotes no detectable damage, whereas a score of 4 reflects severe PT damage. The cumulative distribution curves and the percentage for each damage score within the group are shown for each group. For all significant difference values, KW ANOVA with Dunn’s test *p* values are shown. (**C**) Representative kidney injury molecule-1 (KIM-1)-stained kidney sections from rats (scale bar: 50 μm). (**D**) Quantification of KIM-1 abundance using image color deconvolution (n = 15 slides/group (3 slides per rat), two-way ANOVA with Holm–Sidak; weight (factor1): *p* < 0.0001; treatment (factor2): ns). (**E**) Representative renal cortex images for immunofluorescence staining of TGF-β1 (scale bar: 50 μm) (**F**) Quantification of TGF-β1 abundance using image color deconvolution (n ≥ 9 slides/group, two-way ANOVA with Holm–Sidak; weight (factor1): *p* < 0.04; treatment (factor2): ns). CTRL—control group; CTRL + ML—lean rats with melatonin administration; OB—obese group, OB + ML—obese rats with melatonin administration; ns indicates no statistically significant difference.

**Figure 6 biomolecules-16-00036-f006:**
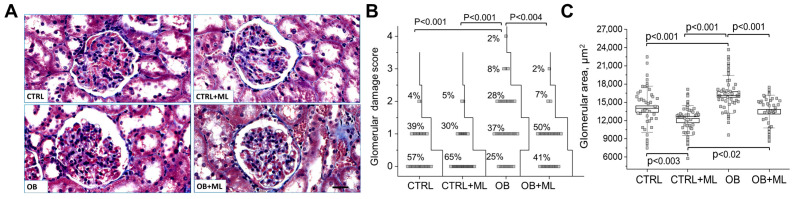
Melatonin attenuates glomerular injury. Assessment of glomerular damage. (**A**) Representative Masson-stained kidney sections from rats (scale bar: 50 μm) for glomerular damage scoring (score 1–4). (**B**) Glomerular damage scores (0–4) are presented for the control and experimental groups. A score of 0 denotes no detectable damage, whereas a score of 4 reflects severe glomerular injury. The cumulative distribution curves and the percentage for each damage score within the group are shown for each group. For all significant difference values, KW ANOVA with Dunn’s test *p* values are shown (Cramer’s V 0.380 for OB vs. OB + ML group). (**C**) Quantification of glomerular cross-sectional area (n = 15 slides/group (3 slides per rat), two-way ANOVA with Holm–Sidak; weight (factor1): *p* < 0.0001; treatment (factor2): *p* < 0.0001). CTRL—control group; CTRL + ML—lean rats with melatonin administration; OB—obese group; OB + ML—obese rats with melatonin administration; ns indicates no statistically significant difference.

**Figure 7 biomolecules-16-00036-f007:**
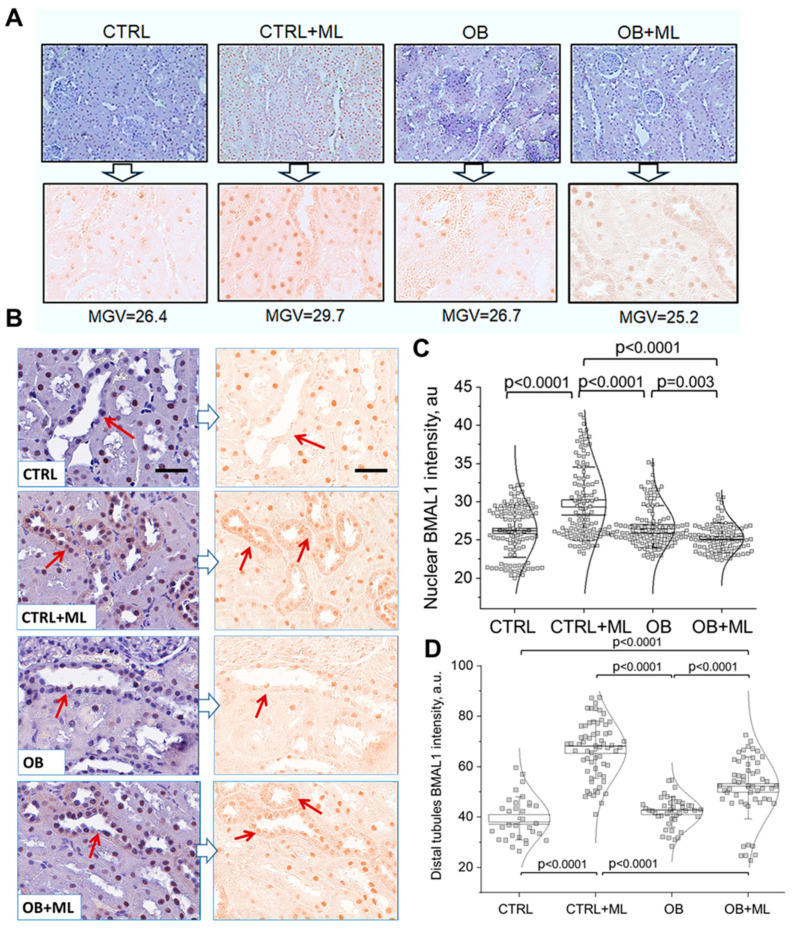
Melatonin alters BMAL1 abundance in the kidney. (**A**) Images from BMAL1-stained slides were taken at 20× using Nikon Ti-2 microscope equipped with NIS-Elements AR software. Then, images were loaded to FIJI and batch-processed for color deconvolution using Macro plugin, and H-DAB staining was analyzed for intensity (mean gray value (MGV) parameter). (**B**) Representative BMAL1-stained kidney cortex sections from rats (scale bar: 50 μm). In addition to nucleus localization, melatonin-treated groups show cytoplasmic presence of BMAL-1 in the distal tubules (DT, arrow). (**C**) Quantification of BMAL1 staining intensity in kidney cortex (n = 15 slides/group (3 slides per rat), two-way ANOVA with Holm–Sidak; weight (factor1): *p* < 0.0001; treatment (factor2): *p* < 0.0001). (**D**) Quantification of BMAL1 staining intensity in distal tubules (n = 15 slides/group (3 slides per rat), two-way ANOVA with Holm–Sidak; weight (factor1): *p* < 0.0001; treatment (factor2): *p* < 0.0001). CTRL—control group; CTRL + ML—lean rats with melatonin administration; OB—obese group; OB + ML—obese rats with melatonin administration; ns indicates no statistically significant difference.

## Data Availability

The original contributions presented in this study are included in the article/Appendix A. Further inquiries can be directed to the corresponding author; raw data will be made available on publicly accessible servers per AHA Data Sharing Policy by 12/2029.

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
