# Peer review of "Melatonin Administration Attenuates High-Fat-Diet-Induced Renal Damage in Wistar Rats"

_biomolecules, 2025, doi:10.3390/biom16010036_

Round 1

Reviewer 1 Report

Comments and Suggestions for Authors
  • Introduction is heavy on non-specific, non-directional descriptions of the background and conclusion, e.g. line 73-76 (MT1 and MT2 receptors in SCN neurons modulate intracellular signaling pathways which were shown to influence the transcriptional-translational feedback loop…), line 92 (in CKD night secretion of melatonin is significantly impaired…), line 99 (disruption of BMAL1 expression has been linked to…), line 101 (melatonin modulates BMAL1 expression and activity, thereby contributing to the maintenance of metabolic homeostasis). Using a language which is more direct and defines the direction of discussed changes or alterations would significantly improve the manuscript.
  • Background section deserves revision. It reads as individual paragraphs presenting abundant number of facts and information, rather than easy to follow, coherent story.
  • As food and water intake were monitored daily, please provide include these data (weekly average values would be sufficient; the same as on Fig 3A). Since reduction in BW gain was observed 2 weeks after ML initiation it is very important to provide the evidence that the observed changes are due to ML action itself and not due to lesser HFD consumption.
  • Any there any functional data/readouts on HFD-induced kidney injury and protective effect of ML available to support histological observations?
  • PSR staining at the level of 12-15% seems very high relative to representative pictures (Fig4C). It would be valuable validation to test for other markers (e.g. TGFb, Pro-C3, Pro-C6, MMPs, TNFRs, MPC-1).
  • Based on Fig 5A and Fig 6A CRL+ML presents with moderate/severe tubular and glomerular injury. Discussion of those results would be a great addition to the manuscript.
  • There is inconsistency between glomerular damage score quantification (Fig 6B) and reporting of the results. Obese rats did not display a significantly higher cumulative glomerular score (p=0.440). Also, ML treatment did not reduce glomerular injury (p=0.348). These results should be revised.
  • There is a lack of connection between HFD-induced kidney injury, ML, and circadian component of the study. Revision is adviced.
  • The discussion is a little bit redundant and mainly contains repeats of background information or ML treatment potential unrelated to presented study. Major revision towards presenting the results in the perspective of current knowledge and other investigator studies is highly recommended.
  • Conclusions from the study require revision. Without additional inflammatory panel data on resistin do not grant the conclusion that ML reduces inflammation. The same with regards to circadian rhythm regulated pathways.

Minor:

  • Figure 1 would be a great supplementary material. What is the expression of ML receptors rat renal tissue?
  • Line 271 – should be “obese group” not “groups”
  • Details on how visceral adipose tissue mass data were calculated and normalized are missing. Changes in adipose tissue mass are relative to what?
  • Since there were body weight differences remove kidney weight [mg] graph, ration of KW/BW is more informative.
  • Y axes for Fig 3D says “related visceral adipose tissue weight, %”. Should it be “relative”?
  • Fig 5C is missing CRTL, CRTL+ML, OB+ML images.
  • Fig 7A and B, OB+ML group image includes a glomerulus while other groups do not. For consistency replace the image for OB+ML to match all the other groups.
  • Fig 7C – add “nuclear” to Y axes label.
  • Materials and methods section is very detailed and well written. The information on images deconvolution (data on Fig 7) would nicely fit in methods section rather than results.

Author Response

We are thankful for the Reviewers' constructive comments and suggestions for our research manuscript “Melatonin Administration Attenuates High Fat Diet-Induced Renal Damage in Wistar Rats" (biomolecules-3943013). We have revised the manuscript to incorporate the requested changes. A detailed description of the changes made is shown below – please find our responses in bold. We hope the revised manuscript meets the high standards required for publication in Biomolecules.

Reviewer #1:

  1. Introduction is heavy on non-specific, non-directional descriptions of the background and conclusion, e.g. line 73-76 (MT1 and MT2 receptors in SCN neurons modulate intracellular signaling pathways which were shown to influence the transcriptional-translational feedback loop…), line 92 (in CKD night secretion of melatonin is significantly impaired…), line 99 (disruption of BMAL1 expression has been linked to…), line 101 (melatonin modulates BMAL1 expression and activity, thereby contributing to the maintenance of metabolic homeostasis). Using a language which is more direct and defines the direction of discussed changes or alterations would significantly improve the manuscript.
  2. Background section deserves revision. It reads as individual paragraphs presenting abundant number of facts and information, rather than easy to follow, coherent story.

Figure 1 would be a great supplementary material. What is the expression of ML receptors rat renal tissue?

Thank you for this helpful suggestion. We significantly revised the Introduction to use more direct and mechanistically focused language. We also added a concise overview of melatonin receptor expression in rodent renal tissue, as requested.

  1. As food and water intake were monitored daily, please provide include these data (weekly average values would be sufficient; the same as on Fig 3A). Since reduction in BW gain was observed 2 weeks after ML initiation it is very important to provide the evidence that the observed changes are due to ML action itself and not due to lesser HFD consumption.

We included the weekly averaged caloric and water intake data in the revised manuscript (new Figure 3B and 3C). These data show that melatonin did not alter caloric intake within either lean or obese groups.

  1. Any there any functional data/readouts on HFD-induced kidney injury and protective effect of ML available to support histological observations?
  2. PSR staining at the level of 12-15% seems very high relative to representative pictures (Fig4C). It would be valuable validation to test for other markers (e.g. TGFb, Pro-C3, Pro-C6, MMPs, TNFRs, MPC-1).

Thank you for this valuable suggestion. We agree and corrected the PSR quantification that was due to a technical thresholding error in the macros. We added additional inflammatory and injury-related markers (TNFR1, TNFα, and TGFβ) to strengthen the functional interpretation of the histological findings (see revised Figures 4B–D).

  1. Based on Fig 5A and Fig 6A CRL+ML presents with moderate/severe tubular and glomerular injury. Discussion of those results would be a great addition to the manuscript.

We updated the figures to ensure the representative images are consistent with the quantitative dataset.  

  1. There is inconsistency between glomerular damage score quantification (Fig 6B) and reporting of the results. Obese rats did not display a significantly higher cumulative glomerular score (p=0.440). Also, ML treatment did not reduce glomerular injury (p=0.348). These results should be revised.

We apologize for the confusion. The values for ordinal data scores showed p value and Cramer V association strength. We improved the clarity of all scoring data and the statistical reporting. The revised figures now display Kruskal–Wallis ANOVA p-values, and Cramer’s V is shown only for assessing association strength within obese groups. Figure legends were also updated for clarity.

  1. There is a lack of connection between HFD-induced kidney injury, ML, and circadian component of the study. Revision is adviced.

We revised the Discussion and Conclusions to more clearly articulate the mechanistic links between obesity-induced renal injury, melatonin treatment, and BMAL1/circadian regulation.

  1. The discussion is a little bit redundant and mainly contains repeats of background information or ML treatment potential unrelated to presented study. Major revision towards presenting the results in the perspective of current knowledge and other investigator studies is highly recommended.

We thoroughly revised and streamlined the Discussion to remove redundancy and better integrate our findings with current literature, as recommended.

  1. Conclusions from the study require revision. Without additional inflammatory panel data on resistin do not grant the conclusion that ML reduces inflammation. The same with regards to circadian rhythm regulated pathways.

We modified the Conclusions accordingly and incorporated additional inflammatory and circadian-related data to support our statements.

Minor:

  • Line 271 – should be “obese group” not “groups”

Corrected.

  • Details on how visceral adipose tissue mass data were calculated and normalized are missing. Changes in adipose tissue mass are relative to what?

We added the requested clarification; please see Lines 181–182 in the revised manuscript.

  • Since there were body weight differences remove kidney weight [mg] graph, ration of KW/BW is more informative.

The kidney weight graph has been removed, and only the KW/BW ratio is presented, as suggested.

  • Y axes for Fig 3D says “related visceral adipose tissue weight, %”. Should it be “relative”?

Thank you for noting this. The label has been corrected to “relative.”

  • Fig 5C is missing CRTL, CRTL+ML, OB+ML images.

This figure has been removed to avoid confusion.

  • Fig 7A and B, OB+ML group image includes a glomerulus while other groups do not. For consistency replace the image for OB+ML to match all the other groups.

The images have been updated for consistency across groups.

  • Fig 7C – add “nuclear” to Y axes label.

Corrected as requested.

  • Materials and methods section is very detailed and well written. The information on images deconvolution (data on Fig 7) would nicely fit in methods section rather than results.

Thank you for the suggestion. We retained the figure in the Results section but added a corresponding description to the Methods to clearly link the methodology with the presented data.

Reviewer 2 Report

Comments and Suggestions for Authors

The authors evaluated the protective effects of melatonin on obesity-induced kidney injury. While similar studies have been conducted by others using diabetic kidney disease as a model, this study links particularly to a circadian marker BMAL1 and showed that BMAL1 exhibits nephron section dependent expression that is related to obesity kidney injury. The study was well-designed. However, the following, if addressed, could improve the manuscript.

Fig. 5B and Fig. 6B, whereby the percentage values are presented in a nonlinear way; I think the presentation is confusing. For example, in Fig. 6 B, for the CTRL, what does 5&%, 39%, and 4% mean? Are the representing damage scores? If so, how come the control already inflicts damage.  Please make the graph straightforward so that there is no second-guessing from readers.

Author Response

We are thankful for the Reviewers' constructive comments and suggestions for our research manuscript “Melatonin Administration Attenuates High Fat Diet-Induced Renal Damage in Wistar Rats" (biomolecules-3943013). We have revised the manuscript to incorporate the requested changes. A detailed description of the changes made is shown below – please find our responses in bold. We hope the revised manuscript meets the high standards required for publication in Biomolecules.

Reviewer #2:

The authors evaluated the protective effects of melatonin on obesity-induced kidney injury. While similar studies have been conducted by others using diabetic kidney disease as a model, this study links particularly to a circadian marker BMAL1 and showed that BMAL1 exhibits nephron section dependent expression that is related to obesity kidney injury. The study was well-designed. However, the following, if addressed, could improve the manuscript.

  1. 5B and Fig. 6B, whereby the percentage values are presented in a nonlinear way; I think the presentation is confusing. For example, in Fig. 6 B, for the CTRL, what does 5&%, 39%, and 4% mean?

We apologize for the confusion. These numbers represent the percentage distribution for each damage score within a group. We clarified this in the figure legends to ensure more intuitive interpretation.

  1. Are the representing damage scores? If so, how come the control already inflicts damage. Please make the graph straightforward so that there is no second-guessing from readers.

Thank you for this thoughtful comment. Yes, the values shown represent renal damage scores based on a standard semiquantitative scoring system previously described in the literature. As the reviewer correctly notes, even healthy control kidneys may exhibit occasional mild tubular changes, since the kidney is a highly dynamic organ that continually adapts to metabolic, hemodynamic, and environmental fluctuations. Scores of 0-1 are considered within the range of normal physiology, whereas scores 3-4 indicate the presence of meaningful tubular injury. We have revised the figure and legend to clarify this scoring framework so that readers can interpret the data without ambiguity.

Round 2

Reviewer 1 Report

Comments and Suggestions for Authors

Thank you for a thorough revision of the manuscript. All major and minor concerns have been nicely addressed. 

For publication purposes the authors may consider revising:

  • the order of paragraphs 5 and 6 in the introduction section. Information in lines 113-122 fits better with "mechanisms of melatonin action in renal tissue" paragraph, and information in lines 129-140 fits better with "melatonin is a circadian regulator of metabolism" paragraph.
  • Food and water intake data presented at Fig3 are most likely weekly not daily averages, but the y axis and figure legend say "daily". 

Author Response

We thank the Reviewers' for their constructive feedback for our research manuscript “Melatonin Administration Attenuates High Fat Diet-Induced Renal Damage in Wistar Rats" (biomolecules-3943013). We have revised the manuscript to incorporate the requested changes. A detailed description of the changes made is shown below – please find our responses in bold. We hope the revised manuscript meets the high standards required for publication in Biomolecules.

Reviewer 1

  1. the order of paragraphs 5 and 6 in the introduction section. Information in lines 113-122 fits better with "mechanisms of melatonin action in renal tissue" paragraph, and information in lines 129-140 fits better with "melatonin is a circadian regulator of metabolism" paragraph.

We appreciate the Reviewer’s thoughtful recommendation. The Introduction has been revised.

  1. Food and water intake data presented at Fig3 are most likely weekly not daily averages, but the y axis and figure legend say "daily".

Thank you. The legend has been corrected accordingly.